# Comparison of Stir Bar Sorptive Extraction and Solid Phase Microextraction of Volatile and Semi-Volatile Metabolite Profile of *Staphylococcus aureus*

**DOI:** 10.3390/molecules25010055

**Published:** 2019-12-23

**Authors:** Kevin Berrou, Catherine Dunyach-Remy, Jean-Philippe Lavigne, Benoit Roig, Axelle Cadiere

**Affiliations:** 1EA7352 CHROME, University of Nimes, Rue du Dr G. Salan, 30021 Nimes CEDEX 1, France; kevin.berrou@unimes.fr (K.B.); benoit.roig@unimes.fr (B.R.); 2Institut National de la Santé et de la Recherche Médicale, U1047, Université Montpellier, UFR de Médecine, 30908 Nîmes, France; catherine.remy@chu-nimes.fr (C.D.-R.); jean.philippe.lavigne@chu-nimes.fr (J.-P.L.); 3Department of Microbiology, CHU Nîmes, Université Montpellier, 30029 Nîmes, France

**Keywords:** gas-chromatography–mass-spectrometry, stir bar sorptive extraction, solid phase microextraction, *Staphylococcus aureus*, bacterial metabolite analysis

## Abstract

For the analysis of volatile bacterial compounds, solid phase microextraction (SPME) is currently the most widely used metabolite concentration technique. Recently, the potential of stir bar sorptive extraction (SBSE) for this use has been demonstrated. These two approaches were therefore used in combination with gas-chromatography coupled with mass-spectrometry (GC–MS) for the analysis of volatile and semi-volatile bacterial compounds produced by *Staphylococcus aureus*. In both cases, SPME and SBSE/headspace sorptive extraction (HSSE) enrichment was carried out in two coating phases. A whole analytical and statistical process was developed to differentiate the metabolites produced from the metabolites consumed. The results obtained with SBSE/HSSE and SPME were compared and showed the recovery of 90% of the compounds by SBSE/HSSE. In addition, we were able to detect the production of 12 volatile/semi-volatile compounds by *S. aureus*, six of which had never been reported before. The extraction by SBSE/HSSE showed higher concentration capacities and greater sensitivity than SPME concerning bacterial compounds, suggesting that this technique may therefore become the new preferred option for bacterial volatile and semi-volatile compound analysis.

## 1. Introduction

Volatile organic compounds (VOCs) are a major concern in many fields such as environmental and medical diagnosis. More particularly, several diseases impact the metabolic profile of specific cells, and the monitoring of VOCs emitted by different cell types could be used as metabolite biomarkers in diagnostics [1,2,3,4,5] and also contribute in the establishment of clinical diagnosis for many pathologies (e.g., cancers, cystic fibrosis, infections) [3,6,7]. 

Among the VOCs, microbial volatile compounds (MVCs) are found. These are intermediates and end-products of various metabolic pathways [8]. Such metabolites (alone or in mixtures) are involved in specific mediating microbial interactions [9]. Indeed, MVCs are engaged in the communication between microorganisms and their living environment. Some of these MVCs can be specifically emitted by a bacterium.

Whichever of the VOCs considered, they are produced in too small amounts to be detected, which implies the use of concentration techniques. To overcome this drawback, techniques combining extraction and concentration have been developed to increase detection sensitivity. Among them, solid phase microextraction (SPME) [10] is the most commonly used. This non-invasive technique consists of the adsorption of VOCs on a fiber followed by thermal or chemical desorption. In medical diagnosis, this technique has already been used for pharmacokinetic studies [11], for the monitoring of drugs and biomarkers during transplant surgery [12]. Considering the concentrations emitted by bacteria, it is often necessary to go through SPME (frequently used for the detection of bacterial metabolites), either for direct detection from a clinical sample [13,14] or after a bacterial culture step [15,16].

To improve concentration of VOCs, Baltussen et al. [17] have developed a derivative of this technique, called stir bar sorptive extraction (SBSE). This method improved the concentration capacity of VOCs by increasing the amount of adsorbent phase. Recently, we proposed the use of SBSE and headspace sorptive extraction (HSSE) simultaneously for the extraction and concentration of volatile and semi-volatile bacterial metabolites in a complex matrix [18]. The model chosen for this study was *Staphylococcus aureus*, bacterium able to cause a wide range of severe diseases [19,20,21,22,23].

However, none of the currently available investigations was able to accurately determine the metabolic profile of *S. aureus*. Indeed, there are conflicting studies on the identification of compounds produced by bacteria [24], and most of these studies do not attempt to be exhaustive and focus only on a few compounds or families of compounds. Currently, among the main volatile metabolites described in the literature as being produced by this bacterium are found 3-methylbutanal, 3-methylbutanoic acid, ethanol, acetic acid, 3-methylbutan-1-ol, acetaldehyde, nonan-2-one, methanethiol, etc. [24].

Discrepancies between the compounds produced or not detected in the different studies can be explained by several factors, including the analytical methods used. Indeed, although the identification of volatile organic compounds (VOCs) is usually performed by gas chromatography coupled with mass spectrometry (CG–MS) [25], other methodologies can be implemented (e.g., IMS, SIFT–MS, PTR–MS, SESI–MS [26,27]). Furthermore, the number of replicates is variable and too low overall (between two and three replicates) for a reliable response.

The objective of this paper is to apply SPME and SBSE/HSSE techniques to study the MVCs emitted by a *S. aureus* culture by GC–MS analysis. Based on the use of 12 replicates, we compared this new approach (SBSE/HSSE) with SPME, commonly used until now for extraction and concentration. By considering the compounds identified following the use of each of these methods, we were then able to discuss their complementarity as well as their respective contributions to the knowledge of the volatile metabolic profile of *S. aureus*.

## 2. Results

### 2.1. Methodology for Compound Identification

The methodology to tentatively identify MVCs is based on several steps as illustrated in Appendix A. Because of the complexity of the chromatogram of a bacterial culture, a first step consists of a mathematical processing (deconvolution) of each chromatographic peak to identify those corresponding to a co-elution and to draw mass spectra corresponding to each individual peak. At the same time, a comparison of the mass spectra obtained with the National Institute of Standard and Technologies (NIST) and the Wiley databases was performed to give a first list of identified compounds—around 240 compounds per sample. Then, the compounds linked to the decomposition of the adsorbent phases (e.g*.*, compounds containing siloxane and polyethylene glycol units) were removed, resulting in a second list of about 60 compounds. To ensure a robust identification, mass spectrum of each compound was checked with databases, the linear retention index (LRI) was calculated and compared with the LRI of the literature (NIST Chemistry WebBook). Compounds with a maximum relative deviation of no more than 2.5% were retained leading to a third list of 19 potential bacterial metabolites.

Finally, these metabolites were compared with the compounds identified in controlled conditions (Figure 1).

Compounds found exclusively in bacterial culture (1-methyl-1-propylhydrazine, 3-ethyl-2,5-dimethylpyrazine, 3-methylbutanoic acid, and acetamide) were considered as being produced by *S. aureus*.

For compounds found under both conditions, a statistical analysis of the peaks areas was performed. Area distribution was assessed using boxplots and outliers were eliminated (Appendix A). The variation of area was observed to determine if the compound was produced (observed area higher than the area of the medium) or consumed (observed area lower than the area of the medium). The significance of the area variation was assessed by a one-way Anova test with a *p*-value < 0.05 (Appendix A).

Conversely, an average area for bacterial culture lower than control indicated a consumption of the volatile/semi-volatile compound by the bacterium.

### 2.2. Comparison of Extraction Method (SPME and SBSE/HSSE)

The efficiency of SPME fibers and SBSE/HSSE bars to extract MVCs from a bacteria culture were tested.

After a preliminary assay of four SPME fibers on a same bacterial sample, only the polydimethylsiloxane/divinylbenzene (PDMS/DVB) and carboxen/PDMS (CAR/PDMS) fibers were retained due to their wide molecules recovery (91%, data not shown).

For SBSE/HSSE, we previously observed that the simultaneous use of two different stir bars (PDMS + ethylene glycol/silicone (EGS)) improved the extraction of bacterial metabolites [18].

Figure 2 represents the distribution of the number of volatile/semi-volatile compounds extracted in *S. aureus* cultures by SBSE/HSSE or SPME (SPME results combine analyses of both fiber types). The numbers of identified compounds were then compared using a Venn diagram. The name of the compounds of each section of this graphic are indicated.

The SBSE/HSSE method extracted and identified 17 compounds while the SPME method found only four molecules (Figure 2). Two out of the four molecules identified using fibers were also detected by SBSE/HSSE. A total of 19 different metabolites were identified in *S. aureus* cultures, 90% of which were extracted via magnetic stir bars highlighting the benefits of this method.

Our results are in agreement with previous studies showing a better efficiency of stir bars due to the larger volume of phase available on stir bars compared to fibers, thus increasing the extraction capacity of volatile/semi-volatile compounds [17,28].

All these results demonstrate the relevance of SBSE/HSSE for the extraction of bacterial metabolites from liquid cultures compared to SPME.

### 2.3. Validation of the Tentative Identification

As shown in Appendix A, the tentative identification of compounds is based not only on the mass spectra but also on the retention index and, for some compounds, analytical standards. All results are listed in Table 1.

The comparison of calculated LRI with LRI available in the literature (with a maximum variation threshold of 2.5%) confirmed the tentative identification of 17 out of 19 MVCs. Only acetaldehyde and propan-2-one were found outside this threshold with respectively 12.5% and 3.1% deviation. However, the use of analytical standards, which showed a similar retention time, confirmed the presence of these two compounds. The highly volatile nature of these two compounds and the difficulties in analyzing them under GC conditions [29] may explain this greater variation in LRI values.

### 2.4. Statistical Analysis to Determine Compounds Consumed and Produced by S. Aureus

The comparison of the compounds extracted from the bacterial culture and those from the control condition showed that only four were found in the bacterial culture (1-methyl-1-propylhydrazine, 3-ethyl-2,5-dimethylpyrazine, 3-methylbutanoic acid, and acetamide) and thus produced by *S. aureus*. 

For the other 15 compounds detected in both control and bacterial culture, the analysis process defined in Figure 1 was applied. Twelve replicates were sampled and analyzed and an Anova test was performed (Table 1).

Three compounds did not show significant variation, which means they came from the lysogeny broth (LB) medium (4-methylquinoline, isoquinoline-1-carbonitrile and quinoline-4-carbaldehyde).

Some compounds showed significant differences (*p*-values less than 0.05) between the control and the bacterial culture. In particular, four area variations showed a decrease in surface between control and *S. aureus* culture. Associated *p*-values were significant, corresponding to compounds consumed by the bacterium (acetaldehyde, 3-methylbutanal, (methyldisulfanyl)methane, and benzaldehyde).

Other tentatively identified compounds showed a significant positive area variation, illustrating a production of these metabolites. Specifically, eight compounds were identified as being produced by *S. aureus* (propan-2-one, ethanol, (methyltrisulfanyl)methane, acetic acid, formic acid, 2-hydroxybenzaldehyde, 1,3,5,7-Tetraazatricyclo[3.3.1.1^3,7^]decane, and 1H-indole).

For the propan-2-one, a *p*-value was indicated. This compound was found in the control, but only in SBSE/HSSE, while in SPME, it was only detected in bacterial culture.

A total of 12 MCVs was tentatively identified as being produced by *S. aureus*.

## 3. Discussion

The detection of MVCs could be of great interest in medical diagnosis. The extraction of metabolites during the growth phase allows higher concentrations to be obtained although a bacterial culture step is necessary. Notably, the presence of the culture medium constituting a complex matrix often of indefinite chemical composition, is an obstacle to the detection of volatile bacterial metabolites. In addition, the volatile metabolic profile of bacteria includes compounds present at very low concentrations. The detection and identification of these molecules represent a real challenge.

This work focused on statistically separating the different molecules identified in order to preserve only those produced by *S. aureus*. A methodology based on the screening without a priori by GC–MS following the extraction and concentration of compounds by SPME or SBSE/HSSE was developed.

The SPME extraction using PDMS/DVB and CAR/PDMS fibers allowed to extract four volatile metabolites (all already described in the literature), while the SBSE/HSSE extraction using EGS and PDMS twisters^®^ allowed the extraction of 17 volatile and semi-volatile metabolites from a bacterial culture medium. SBSE/HSSE has therefore a better extraction capacity than SPME. We also demonstrated the robustness of the method by analyzing 12 independent replicates and developing a process for the treatment of obtained GC data that has shown its effectiveness in statistically determining the compounds produced by bacteria. The diversity of results between the two techniques clearly shows the need for standardization of analytical methods and the use of replicates to obtain reliable results.

Indeed, since the 1980s, many articles have described advances in the metabolic profile of *S. aureus*. In 2013, Bos et al. [24] compiled a review of 15 studies on volatile bacterial metabolites produced by this bacterium. They identified 111 compounds and classified them into three main groups: produced compounds, not produced compounds, and compounds with conflicting results or showing little evidence. Despite the number of studies, only 10 metabolites were considered as produced by the bacterium, 86 metabolites were still awaiting confirmation and 15 have been specifically researched in *S. aureus* culture and never recovered.

Comparison between the literature and our results is summarized in Figure 3 on the basis of the three identical groups.

Ten compounds tentatively identified by our method have already been described in the literature. Seven of them are in agreement with previous works (3-methylbutanoic acid, ethanol and acetic acid as produced compounds, (methyldisulfanyl)methane as not produced, and benzaldehyde, 3-methylbutanal and (methyltrisulfanyl)methane as potentially produced or consumed).

The three other compounds did not match with the literature. Regarding acetaldehyde, its highly volatile nature makes it difficult to detect and quantify under these conditions [29]. So, this compound is found in our study as consumed by *S. aureus*. In addition, propan-2-one and 1H-indole, that have been described controversially [30,31,32,33,34,35,36], have been found as produced compounds. Contradictory results between studies might be explained by several variables like the different strains of *S. aureus* used, the growth medium and the parameters of cultures, the measurements performed at different time points of growth, and the type of method used for the volatile extraction.

Furthermore, the SBSE/HSSE method proposed in our study detected nine new molecules from *S. aureus* cultures, namely: 1-methyl-1-propylhydrazine, 3-ethyl-2,5-dimethylpyrazine, formic acid, 2-hydroxybenzaldehyde, acetamide, 1,3,5,7-Tetraazatricyclo[3.3.1.1^3,7^]decane, 4-methylquinoline, isoquinoline-1-carbonitrile, and quinoline-4-carbaldehyde. The first six were produced by *S. aureus* while the last three ones were only found in the culture medium.

In summary, the comparison of metabolite found with the current literature highlighted that of the 17 metabolites tentatively identified, six have never been reported previously. The SBSE/HSSE extraction method has therefore increased the volatile and semi-volatile metabolic profile of *S. aureus* by providing new information. In addition, the data processing methodology implemented makes it possible not only to accurately identify the volatile compounds, but also to statistically determine whether these compounds were produced, consumed, or simply derived from the culture medium.

In conclusion, SBSE/HSSE can be proposed as a more accurate and complete bacterial volatile and semi-volatile metabolic profile, leading to a greater number of bacterial specific markers. The metabolic profile obtained by SBSE/HSSE represents a powerful approach for the characterization of different strains, which could serve as a decision support system for clinical diagnosis of infectious diseases.

## 4. Materials and Methods

### 4.1. Materials

SPME extractions were performed with four types of fibers: 100 µm polydimethylsiloxane (PDMS)), 65 µm PDMS/Divinylbenzene (PDMS/DVB), 75 µm carboxen/PDMS (CAR/PDMS), and 50/30 µm (DVB/CAR/PDMS), and purchased from Supelco (Supelco Inc., Bellefonte, PA, USA). SBSE/HSSE extractions were performed with 10 mm twister^®^ magnetic stir bars coated with PDMS or ethylene glycol/silicone (EGS) (both 0.5 mm film thickness) obtained from Gerstel GmbH (Mülheim an der Ruhr, Germany). Stir bars and fibers were conditioned prior to use according to the manufacturer′s instructions. In order to control the identification of some compounds, analytical standards were used: ethanol (99.8%) and acetic acid (99.8%) were obtained from VWR (Fontenay-sous-Bois, France). Acetaldehyde (99%), propan-2-one (99.8%), benzaldehyde (99%), and 3-methylbutanoic acid (99%) were purchased from Sigma-Aldrich (Saint-Louis, MO, USA). A standard mixture containing linear alkanes from C7 to C40 from Sigma-Aldrich was used to evaluate the linear retention index.

### 4.2. Culture Conditions

*S. aureus* ATCC 25,923 [37] was the reference strain used in this study. The strain was maintained as frozen stock on Microbank™ bead (Pro-Lab Diagnostic, Richmond Hill, ON, Canada) at −80 °C, and cultivated overnight onto LB medium (tryptone Fluka, Saint-Louis, MO, USA), yeast extract (Amresco, Solon, OH, USA), and sodium salt (Panreac, Barcelona, Spain)) at 37 °C with shaking at 210 rpm. From this culture, 12 replicates were performed for each condition in sterile 20 mL glass vials sealed with PTFE/Silicone screw caps (Agilent, Santa-Clara, CA, USA) containing 10 mL of LB medium, and then was inoculated at 0.1 uDO·mL^−1^. A control with LB medium (12 replicates) was carried out under the same conditions.

### 4.3. Headspace-SPME Extraction Procedure

The vials were incubated in the MultiPurpose Sampler (MPS) agitator for 6 h before extraction at 37 °C with shaking at 250 rpm. Then, the SPME fiber was inserted in the vial headspace and *S. aureus* VOCs were extracted for 30 min at 250 rpm. After the extraction, the fiber was desorbed in the TDU for 90 sec at 220 °C (mean temperature adapted to the four fibers in the preliminary tests) and then, in the subsequent experiments, at 250 °C (PDMS/DVB) or 270 °C (CAR/PDMS) (temperatures recommended by the supplier). Between two extractions, the fiber is reconditioned in the injector for 5 min at the temperature recommended by the supplier.

### 4.4. SBSE/HSSE Extraction Procedure

PDMS and EGS twisters were inserted into liquid cultures of LB medium and in the headspace at the beginning of the experiment for a simultaneous extraction for 6 h of culture at 37 °C and under shaking at 600 rpm. After exposure, the stir bars were removed from the bacterial culture. The PDMS twister was rinsed with ultrapure water and blotted with a lint-free tissue before being placed in a glass thermal desorption tube. The EGS twister was placed directly into the same tube for a simultaneous desorption. In parallel, the procedure was performed (12 replicates) in controlled condition (only LB medium) in order to tentatively identify the compounds from the medium.

### 4.5. GC–MS Analysis

The analytical system was composed of an Agilent 7890 B gas chromatograph coupled with an Agilent 5977 A mass spectrometer with a MPS, a thermal desorption unit (TDU) and a cooled injection system (CIS) (Gerstel). The data acquisition software MSD Chemstation F.01.00 (Agilent Technologies, Les Ulis, France) was used to program the GC-MS. The gas chromatograph was fitted with a VF-WAXms fused silica capillary column (30 m × 0.25 mm × 0.25 µm, Agilent) and was used with helium as carrier gas at 0.8 mL·min^−1^.

The tube was desorbed in the TDU at 220 °C (suitable temperature for both bars, recommended by the supplier) for 5 min. After desorption, VOCs were focalized on the CIS at −10 °C during 2 min, ramped to 250 °C at a heating rate of 12 °C per second, and finally held for 2.5 min in splitless mode to ensure complete desorption of analytes. The column temperature was initially kept at 40 °C for 7 min and then increased from 40 °C to 240 °C at 8 °C·min^−1^, the temperature was maintained during 3 min. Then, the sample was introduced into the ion source of the Agilent 5977 A mass spectrometer. The transfer line temperature was set at 250 °C and ion source temperature at 230 °C. Ions were generated by a 70 eV electron beam. Masses were acquired from *m*/*z* 33–500 amu.

### 4.6. Data Processing and Statistical Analysis

All peaks were integrated using MassHunter Qualitative Analysis software B.06.00 (Agilent Technologies). This software deconvolutes the chromatograms, separating the co-eluted VOCs. Compounds were then identified by their mass spectra by using the NIST and the Wiley7 reference libraries.

The identified compounds present in at least 9 out of 12 replicates were retained for analysis. Compounds found only in the controls were not considered.

The LRI of a compound is the normalization of its retention time on a gas chromatographic column with respect to a homologous series of n-alkanes, thus converting this retention time into a constant independent of the chromatographic system used. For each compound available in the database, the retention index was determined using the one previously obtained on a polar column, as described by Van Den Dool and Kratz [38]. A maximum relative deviation of 2.5% (in the range of the values used in reviews [8,39]) from literature values was accepted to identify bacterial compounds.

Due to the composition of the culture medium, some compounds can be found both in the culture medium alone (control) and in the bacterial culture. It was then necessary to set up a statistical method to discriminate between the compounds produced by the bacteria and those present only in the culture medium. Therefore, the integrated areas of compounds found in each condition form a dataset (between 9 and 12 values) that is first analyzed by boxplots using RStudio software (https://rstudio.com/) to eliminate outliers. Then, the two sets of data corresponding to the same compound present under both conditions were compared by an Anova test, performed by RStudio software.

## Figures and Tables

**Figure 1 molecules-25-00055-f001:**
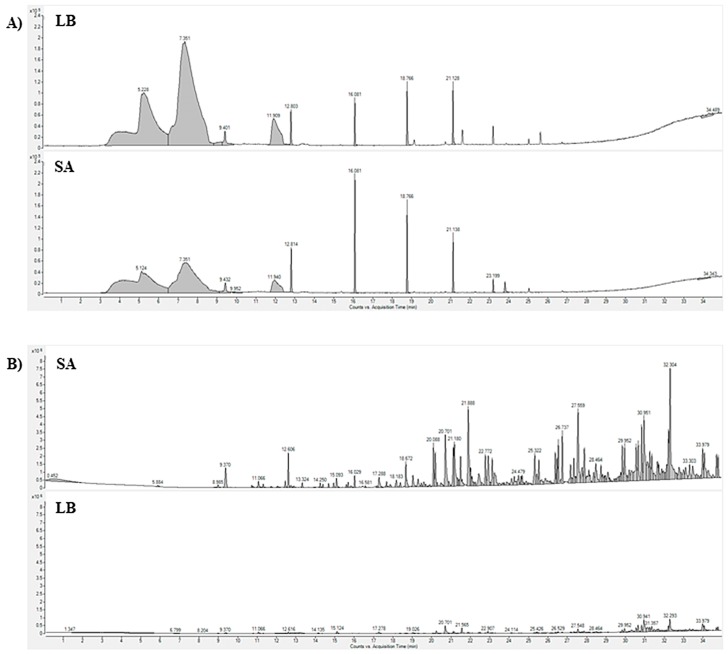
(**A**) Comparison of total ion chromatograms obtained with SPME extraction with CAR/PDMS phase. (**B**) Comparison of total ion chromatograms obtained with SBSE/HSSE extraction. SA: *Staphylococcus aureus* culture condition; LB: control condition; SPME: solid phase microextraction; CAR: carboxen; PDMS: polydimethylsiloxane; SBSE: stir bar sorptive extraction; HSSE: headspace sorptive extraction.

**Figure 2 molecules-25-00055-f002:**
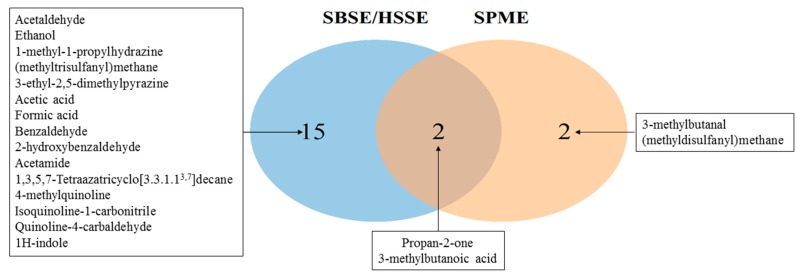
Distribution of the number of volatile/semi-volatile compounds found in *S. aureus* cultures and extracted by SBSE/HSSE or SPME. Venn diagram illustrating the degree of overlap of extracted and identified compounds between the SBSE/HSSE and the SPME methods. The central section in grey represents the compounds that are found by both methods. Specific compounds extracted by SBSE/HSSE are in blue while those extracted by SPME are in orange.

**Figure 3 molecules-25-00055-f003:**
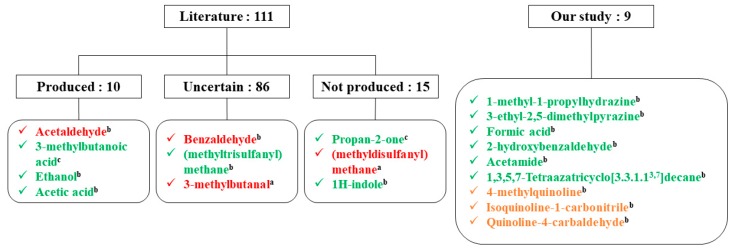
SBSE/HSSE contribution to the detection of volatile/semi-volatile metabolites produced by *S. aureus*. In green: produced compounds; in orange: medium compounds, neither produced nor consumed during growth; and in red: consumed compounds. ^a^ detected by SPME, ^b^ detected by SBSE/HSSE, and ^c^ detected by both methods.

**Table 1 molecules-25-00055-t001:** Compound tentatively identified in a *S. aureus* culture by SBSE/HSSE or SPME-GC-MS and their statistical variation between culture and control under the same conditions.

Volatile/Semi-Volatile Metabolite	Identification			Characterisation		
	LRI ^a^		Percentage ΔLRI	Area Variation	*p*-Value	Produced/Consumed by *S. aureus*
	Measured	Literature				
Acetaldehyde ^c^	803	714	12.5%	↘	3.12 × 10^−3^	Consumed
Propan-2-one ^c^	843	818	3.1%	↗	4.80 × 10^−5^	Produced
3-methylbutanal	866	876	1.1%	↘	6.38 × 10^−5^	Consumed
Ethanol ^c^	938	940	0.2%	↗	5.32 × 10^−6^	Produced
(methyldisulfanyl)methane	1059	1039	1.9%	↘	2.92 × 10^−4^	Consumed
1-methyl-1-propylhydrazine	1066	- ^b^				Produced
(methyltrisulfanyl)methane	1376	1376	0.0%	↗	3.38 × 10^−2^	Produced
3-ethyl-2,5-dimethylpyrazine	1454	1450	0.3%			Produced
Acetic acid ^c^	1464	1479	1.0%	↗	7.54 × 10^−3^	Produced
Formic acid	1531	1510	1.4%	↗	4.89 × 10^−6^	Produced
Benzaldehyde ^c^	1542	1534	0.5%	↘	1.97 × 10^−4^	Consumed
3-methylbutanoic acid ^c^	1645	1655	0.6%			Produced
2-hydroxybenzaldehyde	1702	1680	1.3%	↗	2.41 × 10^−7^	Produced
Acetamide	1767	1773	0.3%			Produced
1,3,5,7-Tetraazatricyclo[3.3.1.1^3,7^]decane	1946	- ^b^		↗	1.88 × 10^−3^	Produced
4-methylquinoline	2152	2108	2.1%	=	6.90 × 10^−1^	No variation
Isoquinoline-1-carbonitrile	2391	- ^b^		=	6.12 × 10^−1^	No variation
Quinoline-4-carbaldehyde	2500	- ^b^		=	5.37 × 10^−1^	No variation
1H-indole	2502	2451	2.1%	↗	2.76 × 10^−2^	Produced

^a^ LRI: Linear retention index. ^b^: LRI not available in the literature. ^c^: Metabolites that have additionally been confirmed by analysis of standards. Percentage ΔLRI: Percentage change between theoretical and calculated values. ↗: Positive area variation of deconvoluted peaks (i.e., mean peak area detected in bacterial culture above that of the control). ↘: Negative area variation of deconvoluted peaks (i.e., mean peak area detected in bacterial culture below that of the control). =: No significant area variation of deconvoluted peaks.

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
