# Peer review of "Comparison of Stir Bar Sorptive Extraction and Solid Phase Microextraction of Volatile and Semi-Volatile Metabolite Profile of Staphylococcus Aureus"

_molecules, 2019, doi:10.3390/molecules25010055_

Round 1

Reviewer 1 Report

The manuscript can be recommended for publication after minor revision. 

Figure 1 should be provided as supplementary material while Figures S1 and S2 shpuld be provided in main document.

The text should be revised with regards to English language and abbreviations should be explained when cited first time.

Author Response

We thank the reviewer for the valuable comments.

Reviewer 2 Report

The current manuscript, authored by K. Berrou and co-workes, assesses the performance of SBSE, as alternative to SPME, for the extraction and identification of volatile organic compounds produced/consumed by microorganism. Cultures of S. aureus were employed through the research for the systematic comparison of both extraction techniques. Moreover, authors describe the identification of new volatile metabolites not reported in previous studies. In general lines, the manuscript is very well structured and written. The main findings of the study are also of interest for the scientific community in between microbiologists and analytical chemists. In summary, authors report that SBSE extraction, based on the use of two Twister coated with different phases, offers richer and most intense GC profiles than SPME. Very likely, the use of a CIS unit during desorption of coated stir-bars improves the efficiency of the chromatographic separation for most volatile species, reducing peak width values for compounds with lower LRIs (see Fig. 1 and 2 in the supplementary section). So, for this section of the chromatogram, more intense peaks do not necessary mean higher extracted amounts of the compounds; anyway compounds detectability has been improved.

My impression of the manuscript is positive and, thus, I recommend its publication; however, moderate revision is required to better define some technical issues in the study.

Remarks:

- The plots shown in Fig. S1 and S2 look more as the raw TIC chromatograms than deconvoluted chromatograms. Please revise. Have authors find any advantage of Mass Hunter software for deconvolution purposes instead of the Unknown Analysis package also from Agilent company? I guess that the 2nd option is much faster than the former.

- Please, specify what is the response variable employed to compare the level of compounds in Table 1 in controls and cultures with S. Aureus? I mean if authors are using the peak areas for deconvoluted compounds, or those measure for an intense ion (molecular ion or fragment ion) from their EI-MS spectra.

- Table 1, Fig. 3. Identification of quinoline-type alkaloids does not seem a relevant finding since they are not related to the metabolism of the bacterium. Their concentrations do not change between controls and cultures of S. aureous.

- A relevant question. Authors are using different microextraction times for each technique: 30 min in case of HSSPME versus 6 h for the combined direct and HSSE with Twisters. Please, justify such difference. For compounds with longer equilibrium times higher responses are expected when extended the sampling period. Have authors some data regarding the kinetics of both extraction techniques?

Author Response

(The authors gave the same response as above.)
